# Biological Actions and Molecular Mechanisms of *Sambucus nigra* L. in Neurodegeneration: A Cell Culture Approach

**DOI:** 10.3390/molecules26164829

**Published:** 2021-08-10

**Authors:** Olga Palomino, Ana García-Aguilar, Adrián González, Carlos Guillén, Manuel Benito, Luis Goya

**Affiliations:** 1Department of Pharmacology, Pharmacognosy and Botany, Faculty of Pharmacy, University Complutense of Madrid, Ciudad Universitaria s/n, 28040 Madrid, Spain; ompalomi@ucm.es (O.P.); ana.garcia.aguilar@ucm.es (A.G.-A.); adrigo09@ucm.es (A.G.); 2Department of Biochemistry and Molecular Biology, Faculty of Pharmacy, University Complutense of Madrid, Ciudad Universitaria s/n, 28040 Madrid, Spain; cguillen@farm.ucm.es (C.G.); mbenito@ucm.es (M.B.); 3Spanish Biomedical Research Centre in Diabetes and Associated Metabolic Disorders (CIBERDEM), Instituto de Salud Carlos III, 28029 Madrid, Spain; 4Department of Metabolism and Nutrition, Institute of Science and Food Technology and Nutrition (ICTAN—CSIC), 28040 Madrid, Spain

**Keywords:** *Sambucus nigra*, neurodegeneration, polyphenols, human neuroblastoma, cell culture, autophagy

## Abstract

*Sambucus nigra* flowers (elderflower) have been widely used in traditional medicine for the relief of early symptoms of common cold. Its chemical composition mainly consists of polyphenolic compounds such as flavonoids, hydroxycinnamic acids, and triterpenes. Although the antioxidant properties of polyphenols are well known, the aim of this study is to assess the antioxidant and protective potentials of *Sambucus nigra* flowers in the human neuroblastoma (SH-SY5Y) cell line using different in vitro approaches. The antioxidant capacity is first evaluated by the oxygen radical absorbance capacity (ORAC) and the free radical scavenging activity (DPPH) methods. Cell viability is assessed by the crystal violet method; furthermore, the intracellular ROS formation (DCFH-DA method) is determined, together with the effect on the cell antioxidant defenses: reduced glutathione (GSH) and antioxidant enzyme activities (GPx, GR). On the other hand, mTORC1 hyperactivation and autophagy blockage have been associated with an increase in the formation of protein aggregates, this promoting the transference and expansion of neurodegenerative diseases. Then, the ability of *Sambucus nigra* flowers in the regulation of mTORC1 signaling activity and the reduction in oxidative stress through the activation of autophagy/mitophagy flux is also examined. In this regard, search for different molecules with a potential inhibitory effect on mTORC1 activation could have multiple positive effects either in the molecular pathogenic events and/or in the progression of several diseases including neurodegenerative ones.

## 1. Introduction

*Sambucus nigra* L., (elderflower) is a well-known herbaceous species of the Caprifoliaceae family spontaneously growing in Europe, West Asia, and North America. Elderflower in the form of herbal tea has been used for the relief of symptoms of common cold, and also as an hypoglycemic, purgative, diuretic and diaphoretic treatment [1,2,3,4].

Elder fruits or juices have also been used in the food area, in the processing of jams or jellies or to produce wine. In contrast, leaves and stems from the plant are considered toxic due to their content in cyanogenic glycosides (i.e., sambunigrin and prunasin) and m-hydroxysubstituted glycosides (i.e., zierin and holocalin) which, after hydrolysis, are able to release cyanide [5].

*S. nigra* flower extracts have shown to exert different biological activities. Izzo et al. [6] confirmed the antibacterial activity of elderflower extracts against both Gram-positive and Gram-negative bacteria; also, isolated compounds from elderflower have shown immunomodulatory activity through an inhibition of macrophage release of pro-inflammatory cytokines, this effect being attributed to the inhibition of activation of NF-kB and phosphatidylinositol-3-kinase (PI3K) [7]. An aqueous extract of elderflower was able to significantly increase glucose uptake, glucose oxidation, and gluconeogenesis in rat abdominal muscle. Pancreatic cells treated with the elderflower extract showed a dose-dependent stimulatory effect on insulin secretion [8]. Elderflower infusion also exerted a diuretic effect in experimental animal that was even superior to the one observed with theophylline (5 mg/kg body weight) [9,10].

The biological properties of elderflower have been related to its high content in polyphenolic compounds which are known for their antioxidant potential activity in both in vitro and in vivo studies: flavonoids, including quercetin, rutin, and kaempferol among others; hydroxycinnamic acids such as chlorogenic (CGA), caffeic (CA), and protocateuchic acids. A large number of studies showed the anti-inflammatory, antiviral, antiallergic, vasoprotective and anti-carcinogenic properties exerted by this kind of natural compounds [5,11], including the beneficial effects of CGA as an antioxidant, hepatoprotective, and hypoglycemic agent.

Regarding neurodegenerative diseases, protein aggregates and dysfunctional/damaged organelles are the most common feature that alter cell homeostasis and trigger neurodegeneration. Compelling evidence indicates that the mammalian target of rapamycin (mTOR) is a central cell growth-regulating kinase that exists in two different protein complexes, mTORC1 and mTORC2, and it is implicated in multiple essential processes including cell growth and protein synthesis in response to nutrients, growth factors, and cellular energy conditions, and the development and function of the nervous system. The induction of mTOR leads to the phosphorylation and activation of many target proteins related to translational machinery, ribosomal biogenesis, and cell growth, including the p70 ribosomal S6 kinase and eukaryotic initiation factor 4E binding protein (4EBP). Moreover, mTORC1 negatively regulates autophagy, which is a lysosomal degradation process of eukaryotic cells for clearing out a broad range of cytotoxic proteins and damaged or dysfunctional organelles, which contributes to maintain cytoplasmic quality control [12]. In this context, stimulating autophagy contributes to neuroprotection by reducing oxidative stress, representing a new therapeutic approach in the prevention and treatment of neurodegenerative diseases [13]. For instance, autophagy-inducing agents such rapamycin, a relatively selective inhibitor of mTORC1, reduce intracellular aggregates of toxic proteins in neurodegenerative diseases, such as β-amyloid levels in Alzheimer’s disease [14], α-synuclein in Parkinson’s disease [15], or huntingtin species in Huntington’s disease [16]. In addition, due to their biological properties, polyphenols including resveratrol, quercetin, and catechins activate autophagy and act as neuroprotective agents in several disease models. In contrast, mTORC1 signaling is highly linked to oxidative stress and brain disorders, including brain tumors and neurodegenerative diseases.

The modulation of mTORC1 signaling by different extracts from elderflower has not been evaluated. Hyperactivation of the mTORC1 pathway is linked to a disrupted clearance of protein aggregates by autophagy during neurodegeneration, so the aqueous extract of elderflower could exert a beneficial effect by preventing neurodegeneration through a significant reduction of mTORC1 signaling on SH-SY5Y cells.

Thus, in this work, a human neuroblastoma (SH-SY5Y) cell line was chosen as a cell culture model of nervous cells and treatment with a strong pro-oxidant, tert-butylhydroperoxide (t-BOOH), was used to reproduce an in vivo condition of oxidative stress in order to study the possible protective mechanisms through which *Sambucus nigra* flower extracts could protect the cell function.

Therefore, the aim of this work was to firstly evaluate the antioxidant potential of different *S. nigra* flowers extracts in relation to their phenolic profile; a human neuroblastoma cell line, SH-SY5Y was used as a cell culture model and in order to study the possible protective mechanisms exerted by a well-characterized extract of *S. nigra* on cell function.

## 2. Results

### 2.1. Antioxidant Activity

In this study, the antioxidant activity of the three different extracts from elderflower were first evaluated by the ORAC assay, which uses Trolox as a water-soluble analogue of vitamin E and is chosen as a positive control in all the assays conducted in this work. Trolox is able to decrease ROS production, to prevent cytotoxicity in human cancer cell lines, and to rescue cells from apoptotic death [17,18]. Then, the oxygen scavenging activity of the samples was determined by the DPPH assay. This activity may be expressed as the inhibitory concentration 50 (IC_50_), which expresses the antioxidant concentration required to obtain 50% radical inhibition, or as the antioxidant efficiency (AE), which is calculated as 1/IC_50_.

Table 1 shows the results for ORAC (as Trolox Equivalents—TE) and DPPH values, which are expressed as IC_50_ and AE for *S. nigra* extracts.

Results indicate that the aqueous and ethanolic extracts exert statistically significant higher antioxidant capacity as measured by the ORAC method when compared to the methanolic one, although their antioxidant ability is moderate. Thus, the following studies will be conducted with the aqueous and ethanolic extracts.

### 2.2. Chemical Profile

The main polyphenols identified in *S. nigra* aqueous and ethanolic extracts by HPLC are listed in Table 2, together with their relative content.

Chromatographic profiles of elder fruit extracts show slight differences in their composition: myricetin is the main polyphenol in the aqueous extract (44%), followed by protocateuchic acid (17%); myricetin is also the main polyphenol in the ethanolic extract (28%), but with a lower percentage and closer to the protocautechic acid content (21%) (Figure 1). These results are in agreement with those previously published [11,19,20,21] ensuring the integrity and quality of the tested samples.

### 2.3. Cell Viability and mTORC1 Signaling

Results from crystal violet assay did not show any cytotoxic effects of aqueous extract from *S. nigra* in SH-SY5Y cells at all the tested concentrations (1–500 µg/mL). However, *S. nigra* methanolic and ethanolic extracts at high concentrations significantly reduced cell viability in a dose-dependent manner. Specifically, methanolic and ethanolic extracts from *S. nigra* significantly decreased cell number at 500 µg/mL and from 50 to 500 µg/mL, respectively (Figure 2A). In all the experiments assessed, cell viability was not significantly affected with respect to the control condition when *S. nigra* extract concentration was from 1 to 25 µg/mL. Accordingly, biological actions and molecular mechanisms of extracts were analyzed using this range of concentrations.

In order to gain better insight into the molecular mechanisms of *S. nigra* flowers in SH-SY5Y cells, the activity of mTORC1 signaling pathway was evaluated by the quantification of the phosphorylation state of threonine-389 and protein levels of its downstream effector p70S6K. The results demonstrated that the aqueous extract of elderflower significantly reduced the phosphorylation of p70 protein compared to control cells, which indicated that mTORC1 signaling is inhibited in response to all the range of concentration assessed. In contrast, methanolic and ethanolic extracts of elderflower reduced or did not affect the activity of the mTORC1 signaling pathway (Figure 2B). Moreover, since rapamycin is a potent inhibitor of mTORC1 activity and thus reduced cell proliferation [22,23,24], we also used this drug as a positive control to inhibit mTORC1 signaling in the presence or absence of the mitochondrial uncoupler carbonyl cyanide *m*-chlorophenyl hydrazine (CCCP) in SH-SY5Y cells (Figure 2C).

### 2.4. ROS Production

Both aqueous and ethanolic extracts were first tested in unstressed cells to check their direct antioxidant potential. Figure 3A shows that 25 and 50 µg/mL of both extracts significantly reduced the steady-state concentration of ROS, indicating that the amount of phenolic compounds in both doses was enough to decrease the basal ROS production in cultured SH-SY5Y. In this assay, cocoa flavonoid epicatechin was used as a positive control for ROS quenching in SH-SY5Y submitted to oxidative stress [25]. When neuroblastoma cells in culture were treated with 100 µM t-BOOH for 21 h or 200 µM t-BOOH for 3 h, the 2-fold increase of intracellular ROS concentration was indicative of a clear situation of oxidative stress (Figure 3B,C). Interestingly, this intense rise of ROS levels was significantly reduced when SH-SY5Y cells were pre-treated with tested concentrations of both extracts for 21 h prior to the oxidative challenge with 200 µM t-BOOH for 3 h (Figure 3B), and in both cases, chemo-protective response against ROS increase was dose-dependent. A similar dose-dependent reduction of t-BOOH-induced ROS over-production was observed when cultured neuroblastoma cells where co-treated for 24 h with 100 µM t-BOOH and either aqueous or ethanol extracts (Figure 3C).

### 2.5. GSH Concentration

When SH-SY5Y cells were submitted to a situation of oxidative stress by the administration of 200 µM t-BOOH for 3 h or 100 µM t-BOOH for 21 h, cell concentration of GSH decreased to around 60% of basal levels (Figure 4A,B). This significant decrease of GSH was partially reversed by a pre-treatment of cells with 25 µg/mL of aqueous extract and with all three doses of the ethanol extract (Figure 4A). Similarly, co-treatment of neuroblastoma cells with 10 or 25 µg/mL of any of the two extracts, aqueous and ethanol, evoked a partial but significant protection against the t-BOOH-induced GSH depletion (Figure 4B). This result unequivocally indicates that the presence in the culture media of the antioxidant compounds contained in both extracts protects SH-SY5Y cells against the loss of reducing power in a situation of oxidative stress. Although Ramiro-Puig et al. [25] did not test the effect of epicatechin on GSH depletion, pre-treatment of SH-SY5Y with 10 µM of the cocoa flavanol resulted in a complete GSH recovery after stress.

### 2.6. GPx Activity

Treatment of SH-SY5Y cells with a challenge of 200 µM t-BOOH for 3 h or 100 µM t-BOOH for 21 h evoked a significant increase in the activity of this antioxidant defense enzyme as a rational response to the induced ROS overproduction (Figure 5A,B). Pre-treatment of neuroblastoma cells with all three concentrations of any of the two extracts for 21 h before the onset of the condition of oxidative stress resulted in a partial although significant reduction of GPx activity to reach values that were between those of controls and stressed cells (Figure 5A). A very similar result was obtained when cells were co-treated for 24 h with tested concentrations of extracts and the pro-oxidant (Figure 5B). Pre-treatment with flavonoid epicatechin at 10 µM evoked a full recovery of GPx activity after the oxidative stress in SH-SY5Y.

### 2.7. GR Activity

Similar to what was observed in GPx assay, the t-BOOH insult provoked a 30% raise in GR activity to recover the increased oxidized glutathione produced by GPx. Contrary to what was observed with GPx, none of the concentrations of any extract was able to reduce the enhanced GR activity when cells were pre-treated before the stress (Figure 5C). However, GR was reversed to control values when neuroblastoma cells were co-treated with the three tested doses of ethanolic extract as well as with 5 and 25 µg/mL of the aqueous extract (Figure 5D). As in the case of GPx, pre-treatment of SH-SY5Y with 10 µM epicatechin reversed GR activity after the stress.

### 2.8. Autophagy

To explore whether the autophagy process is implicated in the protection exerted by these extracts toward an oxidative insult, SH-SY5Y cells were stimulated with t-BOOH, which is an inducer of oxidative stress. Our results indicated that t-BOOH inhibited the phosphorylation of the unc51-like kinase 1 (ULK1) complex at serine 757 mediated by mTOR, indicating an inhibition of the activity of this signaling pathway. In parallel, in response to t-BOOH, there was a slight induction in the protein expression levels of the autophagy marker microtubule-associated proteins 1A/1B light chain 3 (LC3)-II, which is correlated with an enhanced autophagic activity (Figure 6A–C).

Moreover, pre-treatment of SH-SY5Y cells with aqueous or ethanolic extracts (5–25 µg/mL) of elderflower did not significantly affect the phosphorylation of ULK1 at serine 757, neither LC3-II protein expression, suggesting that the autophagy process plays a minor role in the beneficial effects of elderflower (Figure 6A–C).

Since rapamycin is widely used as a potent activator of autophagy [22,24,26], we made use of it as a positive control to stimulate autophagy in the presence or absence of CCCP in SH-SY5Y cells (Figure 6D,E).

## 3. Discussion

Beneficial effects of elderflower have been previously conferred to its polyphenolic content, mainly flavonoids and hydroxycinnamic acids. Their presence in the tested extracts in this work was confirmed, while slight quantitative differences were found between the aqueous and ethanolic extracts. The results of HPLC in our elderflower samples agree with those previously reported [11,19,20,21], this assuring the quality of samples and reliability of the HPLC assay. Furthermore, their antioxidant capacity was evaluated by two different assays, showing a preventive potential against oxidative stress. Due to their phenolic structure, the main bioactive components of *S. nigra*, flavonoids and hydroxycinnamic acids, have a remarkable oxidant scavenging capacity related to the hydrogen-donating ability and the stability of the phenoxyl radicals formed [27]. Our results of the antioxidant capacity in vitro with realistic doses of elderflower extracts seem to support the antioxidant effect observed in cultured cells with similar or comparable doses of other plant extracts also rich in flavonoids and/or hydroxycinnamic acids [28,29,30,31].

The antioxidant capacity reported above together with the known properties of some compounds from *S. nigra* make the aqueous and ethanolic extracts from this plant interesting candidates for cellular chemo-protection, and, to our knowledge, there are no previous data on cell culture-based study testing the antioxidant effects of *S. nigra* extracts on neuroblastoma cells. Although natural phenolics may have potent antioxidant effects in vitro and in vivo, elevated doses of dietary antioxidants may also act as pro-oxidants in cell culture systems and provoke cellular damage [32]. Consequently, the selection of the tentative range of doses used was based both on a literature search and our previous studies with other phenolic extracts. Thus, the authors have reported concentrations ranging from 1 to 25 µM of chlorogenic acid in human blood after coffee consumption [33,34,35]. In previous works, we have shown a chemo-protective effect with 0.5–50 µg/mL of cocoa phenolic extract [25,28,36], 1–40 µg/mL of *Corema album* extract [29], 1–50 µg/mL of green coffee bean extract [30] and 0.5–50 µg/mL of cranberry extract [31] on cultured hepatic cells. Similarly, we have reported chemo-protective activity for 0.5–100 µg/mL *Vochysia rufa* extract [37], 5–25 µg/mL *Silybum marianum* [38], and 2.5–20 µg/mL cocoa extract [39] in cultured endothelial cells. Since these ranges of concentrations, not far from realistic, showed a chemo-protective effect against oxidative stress in different cell lines, a similar range of concentrations was assayed in cultured neuroblastoma cells and their response in oxidative stress conditions as well as their effect on the autophagy process was tested. Firstly, cell viability of SH-SY5Y was examined in the presence of the indicated concentrations of aqueous, methanolic, or ethanolic extracts from elderflower, revealing that the aqueous extract did not exert any cytotoxic effect at any of the concentration tested when compared to high concentrations of methanolic or ethanolic extracts, which reduced cell viability from 35% to 40% at 500 µg/mL. Similar results were obtained with a methanolic extract (400 µg/mL) from *S. nigra* flowers in different cell lines [40]. Nevertheless, this is the first time that an aqueous extract from elderflower shows a protective effect in neuroblastoma cells.

As expected, treatment of cells with 5–50 µg/mL of the two extracts for 24 h produced no significant cell damage and evoked a dose–response reduction in the cellular ROS production. This result agrees with previous reports indicating that plant hydroxycinnamic compounds [29,30,41,42] and/or flavonoids [28,36,39,43,44] are effective scavengers of oxygen radicals in cultured cells. A decreased production of ROS might reflect a reduced intracellular oxidation with a balanced redox status that could confer the cell an advantageous condition to face a potential oxidative insult. Indeed, when cellular ROS generation was intensified by the addition of a potent pro-oxidant such as t-BOOH, cells that were pre-treated or co-treated with aqueous or ethanolic extracts showed a dose-dependent decrease of intracellular ROS. These results suggest that the ROS generated during the period of oxidative stress were more efficiently quenched in SH-SY5Y cells treated with *S. nigra* extracts, surely reducing any ROS-induced cell damage.

Reduced glutathione (GSH), the main non-enzymatic antioxidant defense in the cell, acts as a substrate in glutathione peroxidase-catalyzed detoxification of organic peroxides, reacts with free radicals, and repairs free radical induced damage through electron-transfer reactions. Indeed, the depletion of cellular GSH seems to play a crucial role in apoptotic signaling [45] and, therefore, maintaining GSH concentration above a critical threshold while facing a stressful challenge represents a decisive advantage for cell survival. The dose-dependent recovery of GSH concentration when SH-SY5Y cells were pre- or co-treated with both aqueous and ethanolic *S. nigra* extracts unequivocally indicates an efficient cellular chemo-protection against the oxidative insult. As in the case of ROS levels, similar results have been reported with other plant extracts rich in antioxidant compounds such as flavonoids and hydroxycinnamic acids in different cell types; for instance, *Corema album* [29], green coffee [30], cranberry [31], *Vochysia rufa* [37], *Silybum marianum* [38], and cocoa [39]. However, this is the first time that a protective effect on GSH has been shown by a plant extract in stressed neuroblastoma cells.

The activity of glutathione-dependent enzymes is essential to prevent/reduce the cytotoxicity of ROS. Enhancement of GPx and GR are critical mechanisms of the cell defense system to face ROS overproduction in conditions of oxidative insults [36,46]. An increase in GPx activity faces ROS overproduction at the expense of GSH, which becomes oxidized and is recovered again to GSH by an activated GR activity. Nevertheless, a prompt return of the antioxidant enzyme activities to basal values once the challenge is surpassed will place the cell in a favorable condition to deal with a new oxidative insult. Thus, the fact that most of the pre- and co-treatment conditions of SH-SY5Y cells with aqueous and ethanolic *S. nigra* extracts managed to avert the long-lasting rise in the activities of GPx and GR induced by oxidative stress guaranteed that the cells were in optimum conditions to survive further oxidative challenges. Again, although a similar antioxidant defense response has been previously reported for other plant extracts and different cell types (see above), this crucial defense mechanism for chemo-protection has never been described for neuroblastoma cells submitted to an oxidative stress. Based on a previous report [22] where pre-treatment of SH-SY5Y with cocoa flavanol epicatechin was able to fully recover stress-induced ROS to basal pre-stress values, this same compound was used at 10 µM as a positive control for all pre-treatment assays related to ROS and antioxidant defenses GSH, GPx, and GR. Further to what was reported for ROS values by Ramiro-Puig et al. [25], the results confirm epicatechin as a chemo-protective compound for all tested antioxidant defenses in neuroblastoma cells.

Moreover, we explored the modulation of mTORC1 signaling and hence autophagy activity on cells stimulated with different *S. nigra* extracts and exposed to an oxidative stress induced by t-BOOH. As predicted, in the presence of t-BOOH, ROS levels were increased, inducing oxidative stress and reducing mTORC1 signaling. In addition, since both aqueous and ethanolic extracts from *S. nigra* regulate the antioxidant defense mechanisms necessary to manage the oxidative challenge, the mTORC1 activity assessed by the ULK1 phosphorylation (S757) was not affected when compared to cells stimulated with t-BOOH. These results are in agreement with a previous study that indicates that low levels of ROS stimulate mTORC1 signaling while high doses of ROS decrease mTORC1 activity in vitro and in vivo [47]. In this context, low levels of mTORC1 activity may be necessary to cellular survival and neuronal function. However, a fine balance of mTOR activation is essential to prevent neurodegeneration, since mTOR hyperactivation has been associated with an increase in the phosphorylation of tau protein contributing to the formation of neurofibrillary tangles in Alzheimer’s disease [48].

In addition, in response to oxidative stress, autophagy was mildly induced in cells treated with the aqueous extract from elderflowers when compared to ethanolic extract, indicating that this process is not significantly activated probably due to the antioxidant capacities exerted by both extracts in SH-SY5Y cells. This results partially agree with a very recent work which demonstrated that *S. nigra* fruit extract exerts its effects by triggering autophagy in dysplastic oral keratinocytes [49].

Our results support the hypothesis that the aqueous extract of *S. nigra* promotes neuroprotection, which is in part due to its antioxidant capacity and via the inhibition of mTORC1 and the recovery of autophagy. Ongoing experiments in SH-SY5Y cells will contribute to deepened research into the specific mechanisms of *S. nigra* governing the clearance of pathological protein aggregates that prevents neurodegeneration. In this sense, myricetin, which was the main phenolic compound present in the aqueous extract of *S. nigra*, has been shown to reduce α-synuclein oligomerization preventing Parkinson’s disease [50].

## 4. Materials and Methods

### 4.1. Reagents and Materials

HPLC-grade acetonitrile and water were purchased by Panreac (Madrid, Spain). Standards of kaempferol, myricetin, quercetin, rutin, caffeic, chlorogenic, and protocautechic acids of HPLC-grade were purchased by Sigma-Aldrich (Madrid, Spain). We also acquired CCCP from Sigma (C2759).

Tert-butylhydroperoxide (t-BOOH), glutathione reductase (GR), reduced (GSH) and oxidized (GSSG) glutathione, dichlorofluorescin (DCFH), o-phthaldialdehyde (OPT), nicotine adenine dinucleotide phosphate reduced salt (NADPH), 2,4-dinitrophenylhydrazine (DNPH), H_2_O_2_, 1,1,3,3-tetraethoxypropane (TEP), gentamicin, penicillin G, and streptomycin were purchased from Sigma Chemical Co. (Madrid, Spain), and rapamycin was purchased from Merck (number 553,210). Acetonitrile, methanol of HPLC grade, dimethyl sulfoxide (DMSO) of analytical grade, sodium hydroxide, sodium chloride, di-sodium hydrogen phosphate anhydrous, potassium di-hydrogen phosphate, as well as formic, hydrochloric, perchloric, and sulfuric acids were acquired from Panreac (Barcelona, Spain). Bradford reagent was from BioRad Laboratories S.A. DMEM culture media and fetal bovine serum (FBS) were from Cultek (Madrid, Spain). All other reagents were of analytical quality.

### 4.2. Sample Preparation

Authenticated samples of dried and comminuted *Sambucus nigra* L. flowers were provided from Arkopharma laboratoires, Spain. A voucher specimen was kept at their facilities. Comminuted dried flowers of *S. nigra* were extracted by maceration using water, ethanol and methanol (3 + 72 h) as solvents. The obtained ethanolic and methanolic extracts were filtered, and the solvents were evaporated by using a rotary evaporator at 40 ºC until dryness. The aqueous extract was freeze-dried and kept protected from humidity until use. Extraction yields were 2%, 1.5%, and 5%, respectively. To perform in vitro tests on SHSY5Y cells, once the ethanolic and methanolic extracts were evaporated or freeze-dried in the case of the aqueous extract, all of them were diluted with DMEM medium with a concentration of 1 mg/mL.

### 4.3. High-Pressure Liquid Chromatography (HPLC) Analysis

For chromatographic analysis, a liquid chromatograph Agilent 1100 series with a Spherisorb C_18_ (250 × 4 mm, 5 µm) column was used with a photodiode array detector at 30 °C. The mobile phase consisted of a solvent A (water/formic acid, 99.9:0.1 *v*/*v*) and solvent B (acetonitrile/formic acid 99.9:0.1 *v*/*v*) gradient at flow rate of 0.4 mL/min. The gradient was 0.01–20.00 min 5% B isocratic; 20.01–50.00 min, 5–40% B; 50.01–55.00 min, 40–95% B; and 55.01–60.00 min 95% B isocratic. The injection volume was 20 L. Peaks were monitored at λ = 254, 330, and 305 nm [21].

A sample of each extract was diluted with MeOH/H_2_O (80:20 *v*/*v*) and filtered through a 0.45 µm membrane filter (GMF, Whatman) before HPLC/DAD analysis. The identification and quantification of the main polyphenolic compounds in elderflower was carried out from the retention times and areas under the curve by comparison with authentic standards (caffeic, chlorogenic and protocautechic acids, kaempferol, myricetin, quercetin, and rutin). Data processing was carried out using Clarity Software (Chromatography Station forWindows). All analyses were performed in triplicate and results were expressed as mg/mL of extract.

### 4.4. Antioxidant Capacity and Oxygen Scavenging Activity

ORAC assay. The antioxidant activity was first evaluated in *S. nigra* extracts by the oxygen radical absorbance capacity (ORAC) method [51].

In brief, a sample of Trolox was mixed with fluorescein in a 96-multiwell plate and then, AAPH was added. AAPH was used to generate peroxyl radicals that oxidize fluorescein, causing a decrease in fluorescence (excitation wavelength 485 nm and emission wavelength 528 nm), which is measured every 4 s for 90 min at 37 °C in a multiwell plate reader (FLUOstar OPTIMA fluorimeter, BMG Labtech, Ortenberg, Germany). Eight increasing concentrations of the dried extracts dissolved in MeOH, ranging from 0.01 to 0.5 mg/mL, were assayed. The results calculate the relationship of the areas under the curve between blank and samples and are expressed as micromoles of Trolox equivalents per gram.

DPPH assay. The oxygen scavenging activity of the samples was determined by the DPPH assay [52] in a multiwell plate reader (FLUOstar OPTIMA fluorimeter, BMG LABTECH). Briefly, a stock solution of DPPH of 23 mg/10 mL MeOH was kept at 5 °C until use; then, increasing concentrations of each sample (10 µg/mL, 25 µg/mL, 50 µg/mL, 100 µg/mL, and 200 µg/mL of the extract) were achieved in each well, and absorbance was recorded at 715 nm. Then, the free radical-scavenging activity of each solution was calculated as the percentage of inhibition, and results are expressed as IC_50_ value, which is defined as the concentration of extract (µg/mL) required to scavenge 50% DPPH radicals. The lower the IC_50_ value is, the higher the antioxidant activity.

For both assays, experiments were done in triplicate.

### 4.5. Cell Culture

The human neuroblastoma (SH-SY5Y) cell line was a kind gift from Prof. Ignacio Torres Alemán, Instituto Cajal, CSIC, Madrid, Spain. The cell line was cultured and passaged in Bio-Whittaker DMEM media supplemented with 10% fetal bovine serum. Cells were maintained in a humidified incubator containing 5% CO_2_ and 95% air at 37 °C and grown in DMEM medium supplemented with 10% FBS and 50 mg/L of each of the following antibiotics: gentamicin, penicillin, and streptomycin. The culture medium was changed every other day in order to remove the not adherent and dead cells, and the plates were usually split 1:3 when they reached confluence.

### 4.6. Cell Treatment

SH-SY5Y cells were stimulated with 1–500 µg/mL of aqueous, methanolic, or ethanolic extracts from *S. nigra* dissolved from the 1 mg/mL stock solution in culture medium during 24 h to evaluate cell viability, and 5–25 µg/mL of aqueous or ethanolic extracts during 24 h in order to analyze mTORC1 signaling activity and autophagy in response to an oxidative insult as a negative control (t-BOOH, 200 µM during 30 min). Cells were treated 40 nM rapamycin as a positive control to inhibit mTORC1 signaling in the presence or absence of the mitochondrial uncoupler carbonyl cyanide m-chlorophenyl hydrazine (CCCP) in SH-SY5Y cells.

Different concentrations of aqueous (5–25 µg/mL) and ethanolic (5–25 µg/mL) extracts from *S. nigra*, dissolved from the 1 mg/mL stock solution in culture medium, were added to the cell plates for 24 h to study a direct/basal effect of the compounds (a higher concentration of 50 µg/mL was also tested in ROS assay). In order to evaluate the protective effect of aqueous and ethanolic extracts against an oxidative insult, two different approaches were performed, co-treatment and pre-treatment. In the co-treatment assay, SH-SY5Y cells were simultaneously treated for 24 h with 100 µM t-BOOH plus any of the four different aqueous and ethanolic concentrations; whereas in the pre-treatment assay, cells were first treated with tested doses of aqueous and ethanolic extracts for 21 h, then washed out and submitted to a new media containing 200 µM t-BOOH for 3 h. Cocoa flavonoid epicatechin (10 µM) was used as a positive chemo-protective control for the assays of ROS and antioxidant defenses [25]. Finally, to explore the impact of these extracts in the modulation of autophagy process in response to an oxidative insult, SH-SY5Y cells were first treated with tested doses (5, 10, and 25 µg/mL) of aqueous and ethanolic extracts for 24 h and the last 30 min with 200 µM t-BOOH.

### 4.7. Evaluation of Cell Viability

SH-SY5Y cells were seeded in 24-well plates at a density of 30.000 cells/cm2 in DMEM supplemented with FBS 10%. The following day, cells were treated with different concentrations (1–500 µg/mL) of aqueous, methanolic, and ethanolic extracts from *Sambucus nigra*, dissolved from the 1 mg/mL stock solution in culture medium during 24 h. Then, cells were washed twice with cold PBS and stained with 0.2% violet crystal (*w*/*v*) in 2% ethanol (*v*/*v*) for 10 min. Plates were rinsed with ddH_2_O, dried, and after the addition of 1% sodium dodecyl sulfate (*w*/*v*) and absorbance at 590 nm was determined using FLUOstar OPTIMA fluorimeter (BMG Labtech, Ortenberg, Germany).

### 4.8. Determination of ROS

Cellular ROS were quantified by the DCFH assay using a microplate reader with slight modifications [42]. For the assay, cells were seeded in 24-well plates at a rate of 2 × 10^5^ cells per well and changed to the different aqueous and ethanolic concentrations (5–50 µg/mL the day after. Prior to the end of the assay, 5 µM DCFH was added to the wells for 30 min at 37 °C. Then, cells were washed twice with serum-free medium before multiwell plates were measured in a fluorescent microplate reader at an excitation wavelength of 485 nm and emission wavelength of 530 nm. After being oxidized by intracellular oxidants, DCFH will become dichlorofluorescein (DCF) and emit fluorescence. By quantifying fluorescence over a period of 120 min, a fair estimation of the overall oxygen species generated under the different conditions was obtained. This parameter gives a very good evaluation of the degree of cellular oxidative stress. The assay has been described elsewhere [42].

### 4.9. Determination of Reduced Glutathione (GSH) Concentration

The content of GSH was quantitated by the fluorometric assay described in [53] with some modifications. The method takes advantage of the reaction of GSH with OPT at pH 8.0, although OPT reacts not only with GSH but also with other thiols, such as methionine, cysteine, and N-acetylcysteine, and in comparison to appropriate controls, it permitted a reliable quantification. After the different treatments, the culture medium was removed, and cells (4 × 10^6^) were detached and homogenized by ultrasound with 5% trichloroacetic acid containing 2 mM EDTA. Following centrifugation of cells for 30 min at 1000× *g*, 50 µL of the clear supernatant were transferred to a 96-multiwell plate for the assay. Fluorescence was measured at an excitation wavelength of 345 nm emission wavelength of 425 nm. The results of the samples were referred to those of a standard curve of GSH.

### 4.10. Determination of Glutathione Peroxidase (GPx) and Glutathione Reductase (GR) Activity

For the assay of the GPx and GR activity, treated cells (4 × 10^6^) were suspended in PBS and centrifuged at 300× *g* for 5 min to pellet cells. Cell pellets were resuspended in 20 mM Tris, 5 mM EDTA, and 0.5 mM mercaptoethanol, sonicated, and centrifuged at 3000× *g* for 15 min. Enzyme activities were measured in the supernatants. The determination of GPx activity is based on the oxidation of GSH by GPx, using t-BOOH as a substrate, coupled to the disappearance of NADPH by GR as described in [42] with slight modifications. GR activity was determined by following the decrease in absorbance due to the oxidation of NADPH utilized in the reduction of oxidized glutathione (2006). Protein was measured by using the Bradford reagent.

### 4.11. Western Blotting

SH-SY5Y cells were washed with ice-cold PBS and then lysed in a buffer containing 1% (*v*/*v*) Nonidet P40, 50 mM Tris/HCl, 5 mM EDTA, 5 mM EGTA, 150 mM NaCl, 20 mM NaF, 1 mM phenylmethylsulfonyl fluoride, 10 μg/mL aprotinin, and 2 μg/mL leupeptin (pH 7.5). Cellular debris was pelleted by centrifugation at 15,000× *g* for 15 min at 4 °C, and the resulting supernatants were collected for protein determination. Samples were subjected to SDS/PAGE (8–15% gels), followed by Western blotting and visualization using an ECL Western blotting detection kit (GE Healthcare Bio-Sciences; Madrid, Spain; RPN2106). Densitometric quantification of blots was performed with NIH ImageJ (https://imagej.nih.gov/ij/; access on 15 June 2021).

### 4.12. Antibodies

The following antibodies were obtained from Cell Signaling Technology (Beverly, MA, USA): anti-LC3B #4108, anti-p70S6K #9202, anti-phospho-p70S6K (Thr389), #9205, anti-phospho-ULK1 (Ser 757) (#14202) and anti-ULK1 (#8054). From Sigma-Aldrich, anti-β-actin (A5316) and anti-α-tubulin (T6199) were used.

### 4.13. Statistics

Statistical analysis of data was as follows: prior to analysis, the data were tested for homogeneity of variances by the Levene test; for multiple comparisons, one-way ANOVA was followed by a Bonferroni test when variances were homogeneous or by a Tamhane test when variances were not homogeneous. The level of significance was *p* < 0.05. A SPSS version 23.0 program has been used.

## Figures and Tables

**Figure 1 molecules-26-04829-f001:**
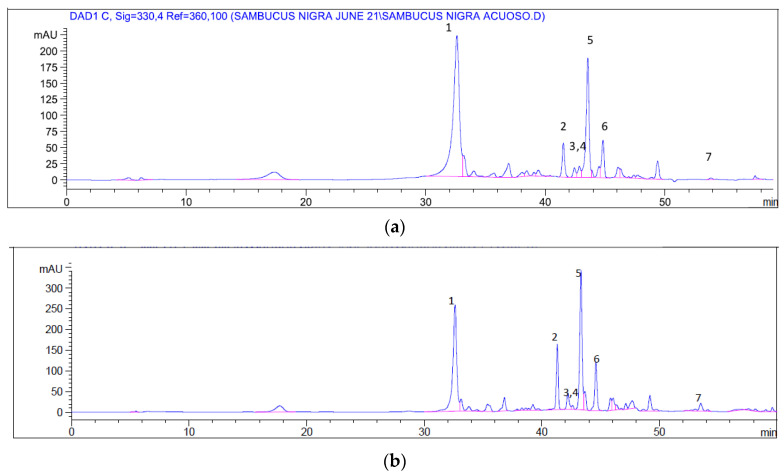
Representative chromatographic profile of (**a**) aqueous and (**b**) ethanolic extracts from *S. nigra* flowers, under the described analytical conditions measured at 330 nm. See identification of peaks in Table 2.

**Figure 2 molecules-26-04829-f002:**
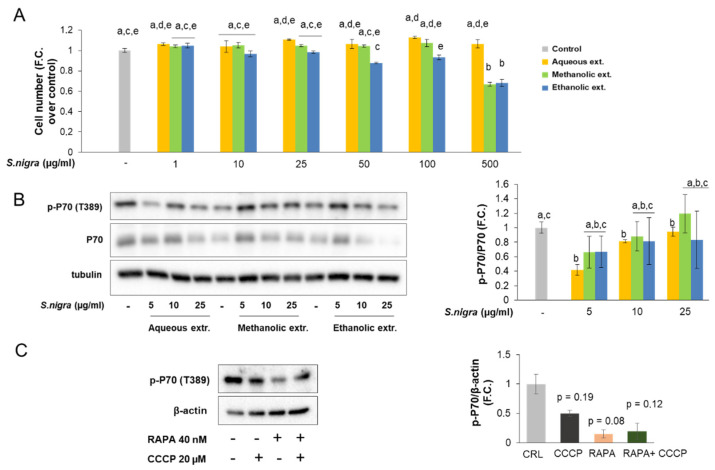
Effect of aqueous, methanolic, and ethanolic extracts from *S. nigra* flowers on cell number (**A**) and mTORC1 signaling (**B**) in SH-SY5Y cells. (**C**) Western blots from SH-SY5Y cells treated with the mitochondrial uncoupler CCCP (20 µM) for 2 h in the presence or absence of 40 nM of rapamycin for 24 h with densitometric quantification of p-p70. We used rapamycin (RAPA) as a positive control to inhibit mTORC1 signaling. Values are means ± SD. *n* = 3. Values are expressed as a fold change (F.C.) relative to control condition. Different letters indicate statistically significant differences (*p* < 0.05) among groups. In panel C, *p* values are indicated compared to the indicated control.

**Figure 3 molecules-26-04829-f003:**
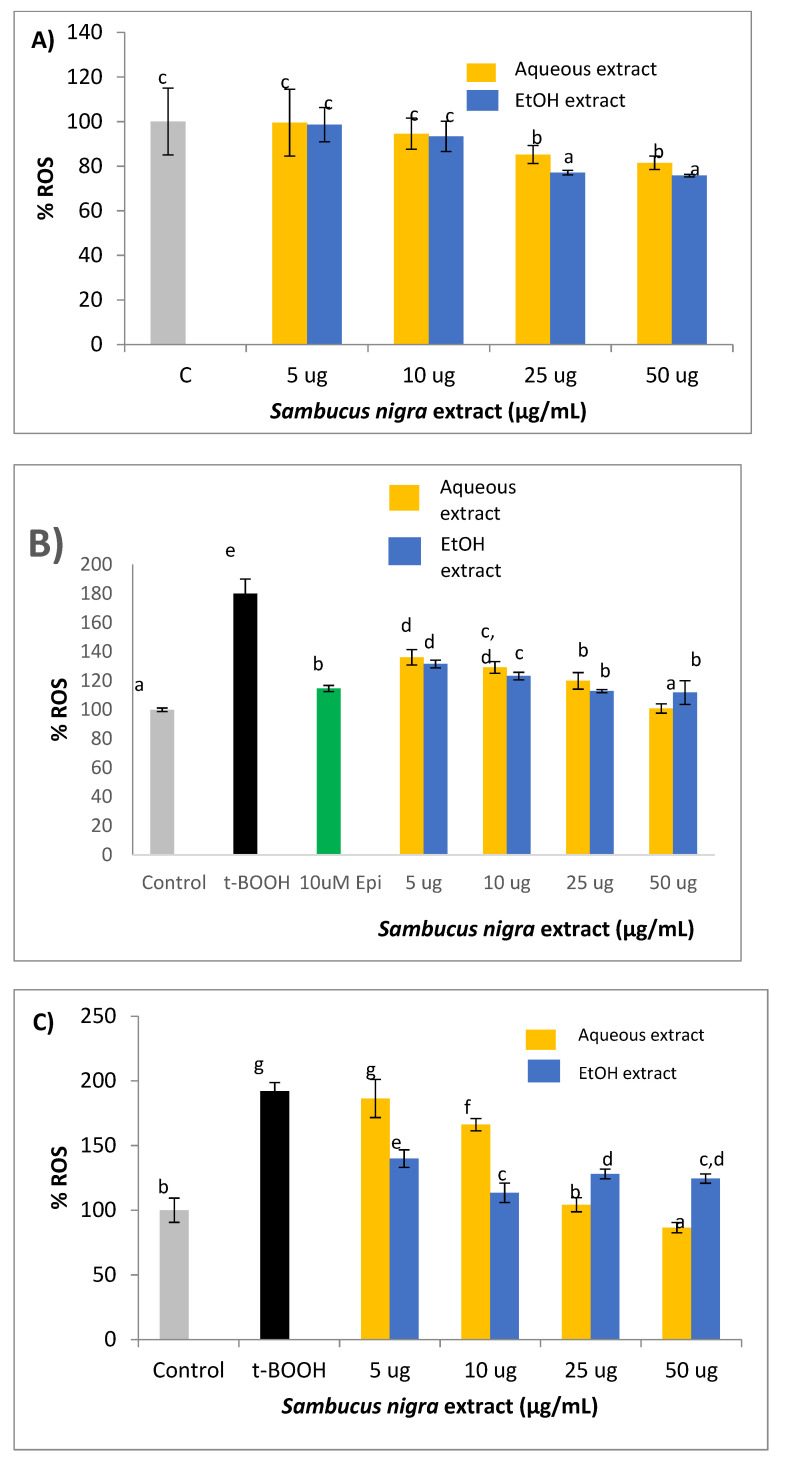
(**A**) Direct effect, (**B**) Pre-treatment, and (**C**) Co-treatment effects of aqueous and ethanolic extracts from *S. nigra* flowers on ROS generation in SH-SY5Y cells. Values are means ± SD, *n* = 4. Values are expressed as a percent relative to control condition. Different letters indicate statistically significant differences (*p* < 0.05) among groups.

**Figure 4 molecules-26-04829-f004:**
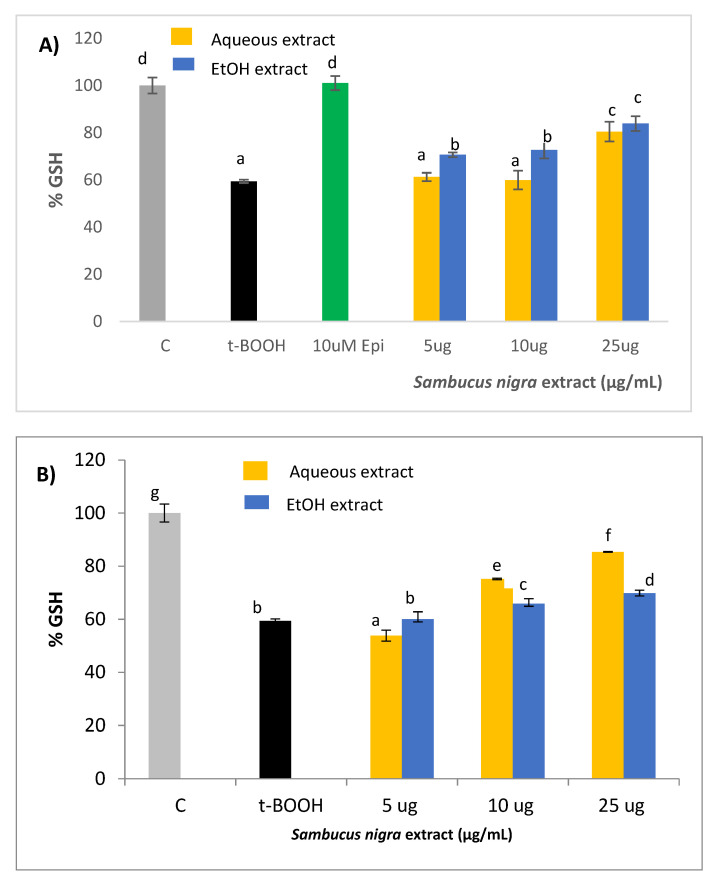
Protective effect of (**A**) Pre-treatment or (**B**) Co-treatment of aqueous and ethanolic extracts from *S. nigra* flowers on GSH levels of SH-SY5Y cells. Values are means ± SD, *n* = 4. Values are expressed as a percent relative to activity of control condition. Different letters indicate statistically significant differences (*p* < 0.05) among groups.

**Figure 5 molecules-26-04829-f005:**
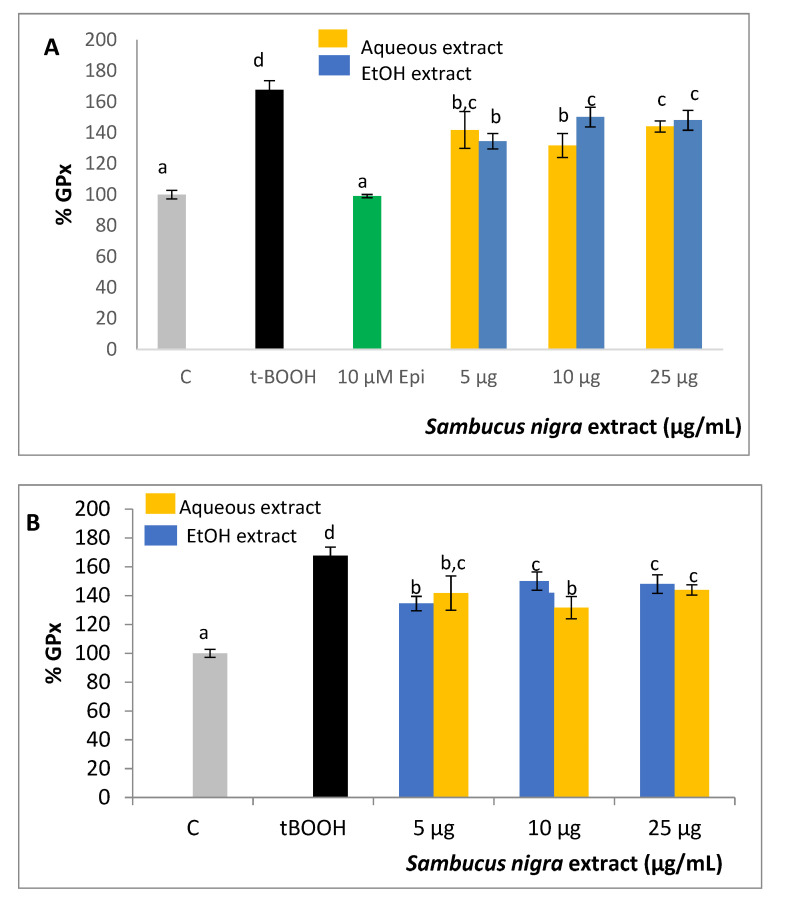
Protective effect of aqueous and ethanolic extracts from *S. nigra* flowers on the enzymatic antioxidant defenses of SH-SY5Y cells. (**A**,**C**) Effect of co-treatment of *S. nigra* extracts on GPx and GR values. (**B**,**D**) Effect of pre-treatment of *S. nigra* extracts on GPx and GR values. Values are expressed as a percent relative to activity of control condition and are means ± SD, *n* = 4. Different letters indicate statistically significant differences (*p* < 0.05) among groups.

**Figure 6 molecules-26-04829-f006:**
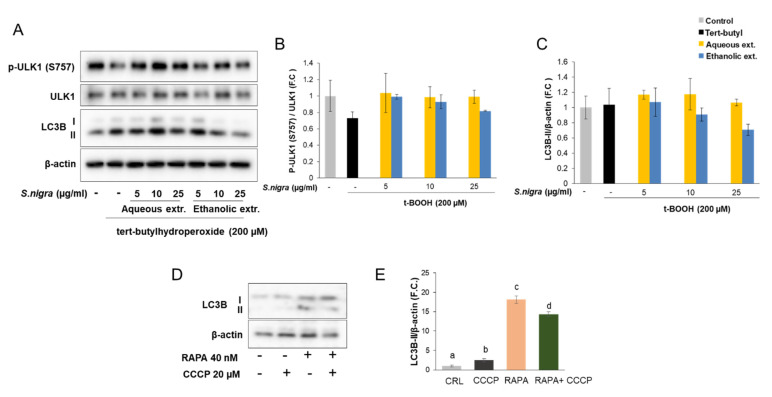
(**A**–**C**) Pre-incubation of SH-SY5Y cells with aqueous or ethanolic extracts from *S. nigra* flowers modulates autophagy in response to oxidative stress induced by tert-butylhydroperoxide treatment (200 µM, 30 min). (**D**,**E**) Western blots from SH-SY5Y cells treated with the mitochondrial uncoupler CCCP for 2 h in the presence or absence of 40 nM of rapamycin for 24 h, with densitometric quantification LC3-II. We used rapamycin (RAPA) as a positive control to stimulate autophagy. Values are means ± SD, *n* = 3. Values are expressed as a percent relative to activity of control condition. Different letters indicate statistically significant differences (*p* < 0.05) among groups.

**Table 1 molecules-26-04829-t001:** ORAC and DPPH values for *S. nigra* extracts. Values are mean ± SD, *n* = 3. Different letters indicate statistically significant differences (*p* < 0.05) among groups.

Extract	ORAC µmol TE/mg	IC_50_ (mg/mL)	AE
Aqueous	1.38 ^b^ ± 0.36	8.17 ^a^ ± 0.30	0.122 ^a^ ± 0.004
Ethanolic	1.13 ^b^ ± 0.28	7.93 ^a^ ± 0.55	0.126 ^a^ ± 0.009
Methanolic	0.66 ^a^ ± 0.03	11.66 ^b^ ± 2.05	0.089 ^b^ ± 0.022

**Table 2 molecules-26-04829-t002:** HPLC profile and content of selected polyphenols from *S. nigra* flowers extracts. Data are mean ± SD, *n* = 3.

Compound	Retention Time (min)	Content (mg/mL Extract)	
		Aqueous Extract	Ethanolic Extract
1 Myricetin	32.61	12.61 × 10^−3^ ± 0.76	9.05 × 10^−3^ ± 0.65
2. Quercetin	40.15	0.45 × 10^−3^ ± 0.05	1.65 × 10^−3^ ± 0.08
3. Caffeic acid	41.48	1.67 × 10^−3^ ± 0.09	0.53 × 10^−3^ ± 0.01
4. Chlorogenic acid	42.42	0.91 × 10^−3^ ± 0.01	0.95 × 10^−3^ ± 0.03
5. Protocateuchic acid	43.56	4.87 × 10^−3^ ± 0.08	7.01 × 10^−3^ ± 0.12
6. Rutin	48.50	0.70 × 10^−3^ ± 0.01	1.12 × 10^−3^ ± 0.04
7. Kaempferol	55.00	0.91 × 10^−3^ ± 0.02	2.50 × 10^−3^ ± 0.11

## Data Availability

Not applicable.

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
