# Peer review of "Biological Actions and Molecular Mechanisms of Sambucus nigra L. in Neurodegeneration: A Cell Culture Approach"

_molecules, 2021, doi:10.3390/molecules26164829_

Round 1

Reviewer 1 Report

While reading the manuscript, I had some questions and recommendations.

  1. Refine the sentence "Elder flower consists of the dried flowers of Sambucus nigra L…" Please provide literature supporting the use of Sambucus nigra flowers as a tea to relieve cold symptoms and as a hypoglycemic, laxative, diuretic and diaphoretic. S. nigra flowers is widely used not only in folk medicine, but also in official medicine, for example, Russia (10.5958/0974-360X.2020.00286.3). Discuss the use of Elder's drugs for the treatment of various diseases.
  2. Please provide the correct reference on the toxicity of various parts of the plant when used as a tea (line 44-45).
  3. Table 1 shows the data for a single definition? Correct the title of column 3-4.
  4. Give the HPLC profile data for the methanol extract.
  5. In table 2, indicate the standard deviation. For the data in the table, provide statistical differences among extracts.
  6. Give an explanation of abbreviations IC50 and EC50.
  7. For better data visualization, present IC50 as mg/ml.
  8. Which reference sample was used as a positive control in in vitro tests (sec. 2.3). Please provide data.
  9. For sections 2.4-2.7, provide the data on the determination of the activity of the methanol extract and the reference sample (positive control).
  10. Specify the country and city for Panreac (HPLC-grade Acetonitrile and Water) and Sigma-Aldrich (kaempferol, myricetin, quercetin, rutin, caffeic, chlorogenic and protocautechic acids of 352 HPLC-grade).
  11. Indicate the time and place of collection of Sambucus nigra L. flowers. Who identified the sample?
  12. Specify the extraction outputs of all three samples (Sec. 4.2).
  13. Indicate the linear range and R2 (sec 4.3).  
  14. Present the data for the determination of phenolic compounds in extracts in terms of the standard substances.    
  15. For multiwell plate reader, please indicate manufacturer, city, and country.
  16. For in vitro tests, provide kit manufacturers and catalog numbers. For all methods used, give in which solvent the extracts and fractions were introduced, the range of investigated concentrations, what were the positive and negative controls.
  17. Conduct a correlation analysis between phenol content and activities to confirm the conclusions. 

Author Response

Dear reviewer,

Thank you for your comments which helped the authors to improve the quality of the manuscript. Please, find our response written in blue color following each comment.

  1. Refine the sentence "Elder flower consists of the dried flowers of Sambucus nigraL…" Please provide literature supporting the use of Sambucus nigra flowers as a tea to relieve cold symptoms and as a hypoglycemic, laxative, diuretic and diaphoretic.  nigra flowers is widely used not only in folk medicine, but also in official medicine, for example, Russia (10.5958/0974-360X.2020.00286.3). Discuss the use of Elder's drugs for the treatment of various diseases. The sentence has been rewritten and references have been included for other medicinal uses: European Medicines Agency Monograph, 2007; Charlebois et al., 2010; Mikulic-Petkovsek et al., 2015; Uncini et al., 2005
  2. Please provide the correct reference on the toxicity of various parts of the plant when used as a tea (line 44-45). The correct reference has been included: Mlynarczyk et al., 2018
  3. Table 1 shows the data for a single definition? Correct the title of column 3-4. Data included in Table 1 are the mean ± SD, n=3. It has been included and the title in columns 3,4 has been corrected. After answering to question 3, this table is now Table 2
  4. Give the HPLC profile data for the methanol extract. In this work, the antioxidant activity of the three different extracts from elder flower (methanolic, ethanolic, aqueous) were first evaluated by the ORAC and DPPH assays. Results shown that the methanolic extract exerted a minor antioxidant ability and thus, the following studies, including phytochemical analysis, were conducted with the aqueous and ethanolic extracts. For more clarity in the manuscript, the HPLC analysis (Chemical profile) has been moved to the second paragraph under section 2. Results.
  5. In table 2, indicate the standard deviation. For the data in the table, provide statistical differences among extracts. Standard deviation and statistical differences have been included. After corrections cited above, this table is numbered as Table 1.
  6. Give an explanation of abbreviations IC50 and EC50. The explanation has been included and EC50 value has been substituted by the correct parameter which is the AE: The inhibitory concentration (IC50) expresses the antioxidant concentration required to obtain 50% radical inhibition; the antioxidant efficiency (AE) is calculated as 1/IC50.
  7. For better data visualization, present IC50 as mg/ml. Data are expressed as mg/mL
  8. Which reference sample was used as a positive control in in vitro tests (sec. 2.3). Please provide data. The authors thank this comment. We incorporated a new panel (panel C) in the new figure 2 in which we used rapamycin (RAPA), an inhibitor of mTORC1, as a positive control to inhibit mTORC1 activity both at basal levels and in response to oxidative stress induced by CCCP stimulation (see new figure 2). So that, rapamycin is extensively used to reduce cell proliferation and to inhibit mTORC1 signaling.

We also incorporated the following references (see section 2.3):

- Lin, X., Han, L., Weng, J., Wang, K., & Chen, T. (2018). Rapamycin inhibits proliferation and induces autophagy in human neuroblastoma cells. Bioscience reports, 38(6), https://doi.org/10.1042/BSR20181822

- García-Aguilar A, Guillén C, Nellist M, Bartolomé A, Benito M. TSC2 N-terminal lysine acetylation status affects to its stability modulating mTORC1 signaling and autophagy. Biochim Biophys Acta. 2016 Nov;1863(11):2658-2667. doi: 10.1016/j.bbamcr.2016.08.006. Epub 2016 Aug 16. PMID: 27542907; Chen, D., Wu, X., Zheng, J., Dai, R., Mo, Z., Munir, F. ... Shan, Y. (2018);

- Dewi FRP, Jiapaer S, Kobayashi A, Hazawa M, Ikliptikawati DK, Hartono, Sabit H, Nakada M, Wong RW. Nucleoporin TPR (translocated promoter region, nuclear basket protein) upregulation alters MTOR-HSF1 trails and suppresses autophagy induction in ependymoma. Autophagy. 2021 Apr;17(4):1001-1012. doi: 10.1080/15548627.2020.1741318. Epub 2020 Mar 24. PMID: 32207633; PMCID: PMC8078762.

  1. For sections 2.4-2.7, provide the data on the determination of the activity of the methanol extract and the reference sample (positive control). As explained in point 4, results shown that the methanolic extract exerted a minor antioxidant ability and thus, the following studies were only conducted with the aqueous and ethanolic extracts
  2. Specify the country and city for Panreac (HPLC-grade Acetonitrile and Water) and Sigma-Aldrich (kaempferol, myricetin, quercetin, rutin, caffeic, chlorogenic and protocautechic acids of 352 HPLC-grade). The country and city have been included (Madrid, Spain)
  3. Indicate the time and place of collection of Sambucus nigra flowers. Who identified the sample? Authentized samples of dried and comminuted Sambucus nigra L. flowers were provided from Arkopharma laboratoires, Spain. A voucher specimen was kept at their facilities for future control.
  4. Specify the extraction outputs of all three samples (Sec. 4.2). Extraction yields were 2%, 1.5% and 5%, respectively.
  5. Indicate the linear range and R2(sec 4.3). The linear range and R2 for each standard have not been included as these data are coming from non-original analysis method and are not considered to be relevant for the present work. As written in the manuscript, the method used was developed by Tundis et al., 2019. (Tundis, R.; Ursino, C.; Bonesi, M.; Loizzo, M. R.; Sicari, V.; Pellicano, T.; Manfredi, I. L.; Figoli, A.; Cassano, A., Flower and Leaf Extracts of Sambucus nigra L.: Application of Membrane Processes to Obtain Fractions with Antioxidant and Antityrosinase Properties. Membranes (Basel) 2019, 9, (10) )
  6. Present the data for the determination of phenolic compounds in extracts in terms of the standard substances.Data are included and expressed as mg/mL of extract.
  7. For multiwell plate reader, please indicate manufacturer, city, and country. A microplate reader FLUOstar OPTIMA (BMG Labtech, Ortenberg, Germany) was used.
  8. For in vitro tests, provide kit manufacturers and catalog numbers. For this work, methods described in quoted references were followed, no commercial kits were used.

For all methods used, give in which solvent the extracts and fractions were introduced, the range of investigated concentrations, what were the positive and negative controls.

As required by the reviewer, a positive control for a complete chemo-protection of cells against an induced oxidative stress has been included. Based on the report by Ramiro-Puig et al. (2009) where pre-treatment of SH-SY5Y with cocoa flavanol epicatechin was able to fully recover stress-induced ROS to basal pre-stress values, we used this same compound at 10 uM as a positive control for all pre-treatment assays related to antioxidant defenses, ROS, GSH, GPx and GR. Results have been included in the pre-treatment data of revised version and, further than what reported for ROS values by Ramiro-Puig and colleagues, show epicatechin as a chemo-protective compound for all tested antioxidant defenses. We strongly believe that very similar results could be expected with epicatechin in a co-treatment assay.

And to determine in vitro tests on SHSY5Y cells, once the ethanolic and methanolic extracts were evaporated or freeze-dried in the case of the aqueous extract, all of them were diluted with DMEM medium with a concentration of 1 mg/mL and used at the indicated concentrations in the manuscript (see sections 4.6 and 4.7). As required by the reviewer, we incorporated new panels in figures 2 (panel C) and 6 (panels D and E) in which we included the effects of rapamycin used as a positive control to inhibit mTORC1 activity and to strongly active autophagy at basal levels and in response of the oxidative stress mediated by CCCP.

Also, new references were incorporated in these sections (see sections 2.3 and 2.8):

-  Lin, X., Han, L., Weng, J., Wang, K., & Chen, T. (2018). Rapamycin inhibits proliferation and induces autophagy in human neuroblastoma cells. Bioscience reports, 38(6), https://doi.org/10.1042/BSR20181822

- García-Aguilar A, Guillén C, Nellist M, Bartolomé A, Benito M. TSC2 N-terminal lysine acetylation status affects to its stability modulating mTORC1 signaling and autophagy. Biochim Biophys Acta. 2016 Nov;1863(11):2658-2667. doi: 10.1016/j.bbamcr.2016.08.006. Epub 2016 Aug 16. PMID: 27542907

- Dewi FRP, Jiapaer S, Kobayashi A, Hazawa M, Ikliptikawati DK, Hartono, Sabit H, Nakada M, Wong RW. Nucleoporin TPR (translocated promoter region, nuclear basket protein) upregulation alters MTOR-HSF1 trails and suppresses autophagy induction in ependymoma. Autophagy. 2021 Apr;17(4):1001-1012. doi: 10.1080/15548627.2020.1741318. Epub 2020 Mar 24. PMID: 32207633; PMCID: PMC8078762

- Bartolomé, A., García-Aguilar, A., Asahara, S. I., Kido, Y., Guillén, C., Pajvani, U. B., & Benito, M. (2017). MTORC1 Regulates both General Autophagy and Mitophagy Induction after Oxidative Phosphorylation Uncoupling. Molecular and cellular biology, 37(23), e00441-17. https://doi.org/10.1128/MCB.00441-17

  1. Conduct a correlation analysis between phenol content and activities to confirm the conclusions.In this work, the total phenol content hasn’t been determined, as this parameter has been published before and authors concluded that the constituents that occur separately in the elder teas could not be responsible for the antioxidant properties (Viapiana et al., 2017). In our study, this correlation analysis hasn’t been performed, as the chemical profile obtained by HPLC analysis (Figure 1) showed only minor differences in the main constituents of the extracts, but not in the total phenol content.

Reviewer 2 Report

A paper entitled “Biological actions and molecular mechanisms of Sambucus nigra L. in neurodegeneration: a cell culture approach” is submitted to Molecules for further reviewing and publication. Generally, this paper is written well, I recommended that this submission is acceptable for publication after revision.

Minor comments

  1. Please reduce the length of sections “Abstract” and “Introduction”. It will let the readers feel boring.
  2. In Table 2, the “50” should be retyped as subscript.
  3. In Figures 3-6, the name for target organism should be typed as italic form.
  4. Typing errors are still found. Please check it carefully throughout the text.

Author Response

Dear reviewer,

Thank you for your comments which helped the authors to improve the quality of the manuscript.

The answer to each comment is written following each point in italics. 

  1. Please reduce the length of sections “Abstract” and “Introduction”. It will let the readers feel boring. Both sections Abstract and Introduction have been reviewed.
  2. In Table 2, the “50” should be retyped as subscript. It has been corrected
  3. In Figures 3-6, the name for target organism should be typed as italic form. It has been corrected
  4. Typing errors are still found. Please check it carefully throughout the text.

The authors thank this comment and the text has been reviewed accordingly.

Round 2

Reviewer 1 Report

S.nigra flowers is widely used not only in folk medicine, but also in official medicine. But reference Mikulic-Petkovsek et al. (2015) is not correct. European Medicines Agency Monograph describes the use of Sambucus nigra L. fructus (EMA / HMPC / 44208/2012). Please use the correct reference (10.5958 / 0974-360X.2020.00286.3).

Myricetin is extracted with methanol (https://doi.org/10.1016/j.mrgentox.2013.03.011 and others). Give the HPLC profile data for the methanol extract.

For the correct conclusion, please, provide your experimental data on the assessment of the activity of myricetin as a positive control. Please give the concentration of the extract solution in terms of myricetin.

For all methods used as validation, give in which solvent the extracts and fractions were introduced, the range of investigated concentrations, what were the positive and negative controls.

Author Response

Dear reviewer, please find below the answer to each of your comments:

- Sambucus nigra flowers is widely used not only in folk medicine, but also in official medicine. But reference Mikulic-Petkovsek et al. (2015) is not correct. European Medicines Agency Monograph describes the use of Sambucus nigra L. fructus (EMA / HMPC / 44208/2012). Please use the correct reference (10.5958 / 0974-360X.2020.00286.3). The reviewer is right and both references have been corrected: The correct reference for Mikulic-Petkovsek et al., is Mikulic-Petkovsek M, Samoticha J, Eler K, Stampar F, Veberic R. Traditional elderflower beverages: a rich source of phenolic compounds with high antioxidant activity. J Agric Food Chem 2015, 63: 1477-1487; the reference to the EMA monograph on S. nigra fructus has been replaced with the correct reference to S. nigra flos (EMA/HMPC/611504/2016)

- Myricetin is extracted with methanol (https://doi.org/10.1016/j.mrgentox.2013.03.011 and others). Give the HPLC profile data for the methanol extract. The reviewer is right; due to its chemical structure, myricetin is soluble in methanol and other authors reported its presence in the methanolic extract of Sambucus nigra L. (Tundis et al., 2019). Nevertheless, in this work, the antioxidant activity of the three different extracts from elder flower (methanolic, ethanolic, aqueous) were first evaluated by the ORAC and DPPH assays and results shown that the methanolic extract exerted a minor antioxidant ability and thus, the following studies, including phytochemical analysis, were conducted with the aqueous and ethanolic extracts. The HPLC profile of the methanolic extract of S. nigra could be performed by our group, but due to the summer vacation period, it couldn’t be done before the second week of September. Moreover, in our opinion, these data wouldn’t give any profit to the article, as they are secondary to the main goal, which is a cell culture approach of the biological actions and molecular mechanisms of Sambucus nigra L.

- For the correct conclusion, please, provide your experimental data on the assessment of the activity of myricetin as a positive control. Please give the concentration of the extract solution in terms of myricetin. After the first revision sent by the reviewer and as required by the reviewer, a positive control for a complete chemo-protection of cells against an induced oxidative stress was included. Based on the report by Ramiro-Puig et al. (2009) where pre-treatment of SH-SY5Y with cocoa flavanol epicatechin was able to fully recover stress-induced ROS to basal pre-stress values, we used this same compound at 10 uM as a positive control for all pre-treatment assays related to antioxidant defenses, ROS, GSH, GPx and GR. If myricetin is to be used as a positive control in our research, the whole experiments should be performed again and in our opinion, the use of myricetin wouldn’t improve the obtained results and moreover, these data wouldn’t give any profit to the article, as they are secondary to the main goals of the work which are 1) the assessment of the antioxidant and protective potentials of Sambucus nigra flowers in the human neuroblastoma (SH-SY5Y) cell line and 2) to examine the ability of Sambucus nigra flowers in the regulation of mTORC1 signaling activity and the reduction in oxidative stress through the activation of autophagy/mitophagy flux.

- For all methods used as validation, give in which solvent the extracts and fractions were introduced, the range of investigated concentrations, what were the positive and negative controls. These data have been included in the manuscript. In detail: 1) To perform in vitro tests on SHSY5Y cells, once the ethanolic and methanolic extracts were evaporated or freeze-dried in the case of the aqueous extract, all of them were diluted with DMEM medium with a concentration of 1 mg/mL (lines 406-409). 2) For HPLC analysis, A sample of each extract was diluted with MeOH/H2O (80:20 v/v) and filtered through a 0.45µm membrane filter (GMF, Whatman) before HPLC/DAD analysis (lines 418-419). 3) For both ORAC and DPPH methods, the plant dry extracts were dissolved in Methanol and Trolox was used as the positive control; no negative control is used as these methods are based in the comparison of the tested sample with the known antioxidant reference. For ORAC assay, eight increasing concentrations of the extracts ranging from 0.01mg/mL to 0.5mg/mL were used; for DPPH assay, increasing concentrations of each extract (10 µg/ml, 25µg/ml, 50µg/ml, 100µg/ml and 200µg/ml) were assayed. 4) For cell treatment, SH-SY5Y cells were stimulated with 1-500 µg/mL of aqueous, methanolic or ethanolic extracts from S. nigra dissolved from the 1 mg/mL stock solution in culture medium and 5-25 µg/mL of aqueous or ethanolic extracts during 24 h in order to analyze mTORC1 signaling activity and autophagy in response to an oxidative insult; t-BOOH was used as a negative control and rapamycin as a positive control to inhibit mTORC1 signaling in the presence or absence of the mitochondrial uncoupler carbonyl cyanide m-chlorophenyl hydrazine (CCCP). 5) For the ROS assay, concentrations of 5 to 50µg/mL of aqueous and ethanolic extracts from S. nigra, dissolved from the 1 mg/mL stock solution in culture medium were tested; t-BOOH was used as a negative control while epicatechin was used as a positive chemo-protective control for the assays of ROS and antioxidant defenses. 6) For GSH, GPx and GR determination, 5 to 25µg/mL of aqueous and ethanolic extracts from S. nigra, dissolved from the 1 mg/mL stock solution in culture medium were tested; t-BOOH was used as a negative control while epicatechin was used as a positive control.